# A Proposed Algorithm Based on Variance to Effectively Estimate Crack Source Localization in Solids

**DOI:** 10.3390/s24186092

**Published:** 2024-09-20

**Authors:** Young-Chul Choi, Byunyoung Chung, Doyun Jung

**Affiliations:** Korea Atomic Energy Research Institute (KAERI), 989-111 Daedeok-daero, Yuseong, Daejeon 305-353, Republic of Korea; cby@kaeri.re.kr (B.C.); jungdoyun@kaeri.re.kr (D.J.)

**Keywords:** acoustic emission, variance, arrival time, moving window, source localization

## Abstract

Acoustic emissions (AEs) are produced by elastic waves generated by damage in solid materials. AE sensors have been widely used in several fields as a promising tool to analyze damage mechanisms such as cracking, dislocation movement, etc. However, accurately determining the location of damage in solids in a non-destructive manner is still challenging. In this paper, we propose a crack wave arrival time determination algorithm that can identify crack waves with low SNRs (signal-to-noise ratios) generated in rocks. The basic idea is that the variances in the crack wave and noise have different characteristics, depending on the size of the moving window. The results can be used to accurately determine the crack source location. The source location is determined by observing where the variance in the crack wave velocities of the true and imaginary crack location reach a minimum. By performing a pencil lead break test using rock samples, it was confirmed that the proposed method could successfully find wave arrival time and crack localization. The proposed algorithm for source localization can be used for evaluating and monitoring damage in tunnels or other underground facilities in real time.

## 1. Introduction

A semi-permanent underground high-level waste disposal repository can be highly stressed by the high temperature of the used nuclear fuel, underground water, and deep geological conditions [1]. Under such conditions, it is very important to maintain real-time monitoring of the repository’s long-term integrity and evaluate the degree of structural damage to ensure the safety of the disposal system. To evaluate the degree of damage, the most important task is the accurate estimation of crack locations. Time-of-arrival differences (TOADs) of the crack waves must be accurately measured to precisely calculate crack locations.

Acoustic emission sensors are used to measure crack waves. Recently, the acoustic emission (AE) technique has been widely used for the real-time structural health monitoring of a structure [2,3,4]. AE has been widely used to evaluate the damage mechanisms of various structures because it is closely related to crack initiation and growth in materials. A method using acoustic emission to monitor concrete structures [5,6,7] and rock slopes is introduced [8].

A large number of discontinuities formed by blasting and excavation exist around a radioactive waste repository. These discontinuities are the main causes of the scattering and dispersion of elastic waves and interference from reflected waves. In such conditions, the crack wave becomes undetectable because of the noise signal. For this reason, to calculate the exact location of cracks, a method is needed to isolate the crack wave from noise signals.

Traditionally, the estimation of source localization is performed through a triangulation method and a circle intersection technique based on the TOAD [9,10]. Recently, a method of calculating TOAD using time–frequency analysis has been studied. Representative time–frequency analysis includes Short-time Fourier transform (STFT), wavelet transform, and Wigner–Ville distribution [11,12,13,14,15]. In the time–frequency domain, the dispersive characteristics of crack waves are well expressed in homogeneous mediums, such as metals, but the disadvantage is that dispersive characteristics do not come out well in a non-homogeneous medium, such as rock. Thus, much research is underway to estimate TOAD and crack location in a rock [16,17].

This paper suggests an algorithm for determining TOADs using a moving window and for estimating a crack’s location from a crack signal with noise in rocks.

## 2. Basic Idea

Figure 1 shows the crack wave generated from cracks in rocks. The crack wave starts slightly before 1.3 msec; however, a noise signal generated from various causes makes it difficult to accurately determine the arrival time of the crack wave. Given the uncertainty in the arrival time of the crack wave, the estimate of the crack localization contains a significant error. For example, an error of 0.1 msec in the arrival time of a crack wave with a velocity of 5000 m/s may result in errors of 0.5 m in the crack localization. To better assess the integrity of structures, such as tunnels or other underground facilities, a method is needed to reduce the uncertainty in crack wave arrival time detection and crack localization.

To more accurately detect the crack wave arrival time from monitoring signals, we first investigated the characteristics of the crack wave and noise signals, employing frequency analysis. Figure 2 shows the results of the frequency analysis (power spectrum) obtained from the crack wave signal. The crack signal obtained by the AE sensor is a non-stationary signal that has a narrow frequency range, while the noise is a stationary signal that has a broad frequency range, as shown in Figure 2. A stationary signal has the characteristic that the ensemble mean and variance values do not change regardless of the signal length. Therefore, if the variance in the crack signal in a noisy environment is calculated according to the length change in the signal, the variance in the noise does not change, and only the variance in the crack signal changes, as explained in Figure 3.

In this paper, we propose a method of estimating the crack wave arrival time using a moving window. The TOAD is estimated by calculating the variance in the measured signal mixed with noise applied to moving windows of different sizes, as shown in Figure 4. The starting point of the crack signal is the point at which the variances in the measured signal change as the window size changes.

## 3. Background Theory for Finding TOAD

As mentioned above, the algorithm determines the variance from the window covering a certain range of signals. Then, an identical method is applied with windows of varying sizes. The theoretical formulas for the proposed method in this chapter are as follows.

The windowed signal $f_{k}(\tau ,\;T)$ of a crack signal $f\left(t\right)$ can be expressed by the following equation:(1)$$f_{k\;}\left(\tau ,\;T\right)=\frac{1}{M\cdot T}\int_{k\tau }^{k\tau +T}W(t-k\tau )\cdot f\left(t\right)dt\;M=\int_{0}^{T}W\left(t\right)dt$$
$$k=0,1,2,\cdot \cdot \cdot ,N-\frac{T}{∆t}$$
where *W*(*t*) is the window function, *T* is the size of the moving window, *N* is the number of window data points, and ∆t is the sampling time.

The variance is as follows:(2)$$\sigma^{2}=E\left[x^{2}\right]-(E{\left[x]\right)}^{2}$$

If the windowed signal is substituted for Equation (2), the variance in the windowed signal can be shown as follows:(3)$$VS\left(k,\;T\right)=\int_{0}^{T}f_{k}^{2}\left(\tau ,\;T\right)\frac{dT}{T}-{\left\{\int_{0}^{T}f_{k}\left(\tau ,\;T\right)\frac{dT}{T}\right\}}^{2}$$

And Equation (3) can be written as Equation (4).
(4)$$VS\left(k,\;T\right)=\frac{1}{M^{2}T^{4}}\left[T\int_{0}^{T}{\left\{\int_{kT}^{k\tau +T}W\left(t-k\tau \right)f\left(t\right)dt\right\}}^{2}d\tau \right.\;\left.-{\left\{\int_{0}^{T}\int_{kt}^{k\tau +T}W\left(t-k\tau \right)f\left(t\right)dtd\tau \right\}}^{2}\right]$$

As shown in Figure 1, a crack signal generated in rock can be expressed as the sum of the background noise and the p-wave signal. If the background noise is assumed to be white noise, its mean is zero, and its variance is $\sigma^{2}$, then Equation (4) can be defined as Equation (5).
(5)$${VS}_{n}\left(k,\;T\right)=\sigma^{2}$$

As shown in Equation (5), the noise signal has a constant value, regardless of the size (T) of the window.

To verify the behavior of the p-wave in Equation (4), the crack wave is considered to be a sine wave with the frequency ω, as follows:(6)$$f\left(t\right)=sin\omega t$$

Substituting Equation (6) into Equation (4) becomes as follows:(7)$$VS\left(k,\;T\right)=\frac{1}{M^{2}T^{4}}\left[T\int_{0}^{T}{\left\{\int_{kT}^{k\tau +T}W\left(t-k\tau \right)sin\left(\omega t\right)dt\right\}}^{2}d\tau \right.\;\left.-{\left\{\int_{0}^{T}\int_{kt}^{k\tau +T}W\left(t-k\tau \right)sin\left(\omega t\right)dtd\tau \right\}}^{2}\right]$$

If a window function $W\left(t\right)$ is assumed to be a rectangular window with a width *T*, it can be written briefly, as shown in the following equation:(8)$${VS}_{p}\left(k,\;T\right)=\frac{1}{M^{2}T^{4}}\left\{T\int_{0}^{T}{sin}^{2}\omega \tau d\tau -{\left(\int_{0}^{T}sin\omega \tau d\tau \right)}^{2}\right\}\;=\frac{1}{4M^{2}T^{4}\omega^{2}}(2\omega^{2}T^{2}-\omega Tsin2\omega T-2cos2\omega T+8cos\omega T-6)$$

As can be seen from Equation (8), now that the crack wave is a function of the size of the window, it can be confirmed that the variance varies according to the size of the window.

Therefore, as confirmed by Equations (5) and (8), even if a crack wave exists in the noise, if the variance is calculated using the proposed method with the moving window, the noise signal does not change, but the variance in the crack wave changes.

The proposed method calculates the variance while changing the size of the window. Here, determining the window size is the most important factor for ascertaining the arrival time of the crack wave. In this section, we will describe how to determine the size of the window.

Figure 5b shows the results of the variance with the size of the window when the crack wave is assumed to be a sine wave. As can be seen from the graph, it can be confirmed that the change in variance is largest within one wavelength of the crack wave. The size of the window should be larger than one wavelength of the crack wave because the proposed method determines the arrival time of the crack wave by observing the change in variance according to the size of the window.

## 4. Source Localization Algorithm

There are several methods for estimating the source location, including a triangular method and a circle intersection method [9,10]. These conventional methods have the disadvantage that source localization is only possible if the velocity of the crack wave is known.

Figure 6 explains the concept of the source localization method without information on the velocity of the crack wave. When a crack signal is produced at its true source, we can observe the signals with different arrival time delays at sensor 1 and sensor 2. The velocities from the image source *x* to each sensor can be expressed as follows:(9)$$V_{1}=\frac{x_{1}-x}{t_{1}-t_{0}},{\;V}_{2}=\frac{x_{2}-x}{t_{2}-t_{0}}\;$$
where $t_{0}$ is the initial crack time and $t_{1},t_{2}$ are the measured delay times at each sensor. Because the velocity of the crack wave is constant, regardless of its source location, the velocities calculated at each sensor are equal. In other words, if the image source moves along the *x*-axis as shown in Figure 6, the true source location is the point at which the velocities calculated at each sensor are equal $\;\left(V_{1}=V_{2}\right)$.

Because the case in Figure 6 is one-dimensional, the true source location can be estimated with just two velocities. In order to estimate the true location for two or three dimensions, we must use three or more sensors. For example, if the experiment uses *N* number of sensors, we can calculate the velocity of the p-wave by selecting two sensors. Therefore, *M* number of velocities are drawn from the following calculation:(10)M=C2 N =N!2N−2!

To define where *M* number of velocities accord with each other, this paper introduces a variance in velocities to calculate the location of the source. The location of an imaginary source, where *M* number of velocity variances become 0, is where *M* number of velocities accord with each other, which is the location of the true source.

In one dimension (Figure 6), the variance in velocities from each sensor is
(11)$$\sigma_{V}^{2}=\left\{{\left(\bar{V}-V_{1}\right)}^{2}+{\left(\bar{V}-V_{2}\right)}^{2}\right\}\;=\frac{1}{2}{\left(V_{1}-V_{2}\right)}^{2}$$
where $\bar{V}=\frac{1}{2}\left(V_{1}+V_{2}\right)$ Then, substitute Formulas (9)–(11).
(12)$$\sigma_{V}^{2}=\frac{1}{2}\frac{{\left(x_{2}-x_{1}\right)}^{4}}{{\left(t_{2}-t_{1}\right)}^{2}{\left(x_{1}-x_{s}\right)}^{2}{\left(x_{2}-x_{s}\right)}^{2}}{\left(x-x_{s}\right)}^{2}$$

In Formula (11), when an image source location accords with a true source location ($x-x_{s}$), the variance is 0. Thus, the true source location is where the variance in the velocities is minimal.

The general formula to define a crack location in three dimensions is
(13)$$\sigma_{V}^{2}=\frac{1}{M-1}\left\{\sum_{n=1}^{M}{\left(\bar{V}-V_{n}\right)}^{2}\right\}$$

In this formula, $\bar{V}=\frac{1}{M}\sum_{n=1}^{M}V_{n}$, and $V_{n}$ means velocity from an imaginary source to the *N*th sensor.

## 5. Validation Using an Experiment

To verify the proposed method for estimating the arrival time and location of the crack signal, an experiment was carried out using six AE sensors on the rock specimens. The AE signal was generated using a pencil lead break and the locations of source points and sensors are shown in Figure 7.

As shown in Figure 8, the AE signal was measured using AE-300, which is a program developed by Rectuson & Fuji ceramics, and the AE-603 SW-GA sensor, which has a resonance frequency band in the range of 60 kHz ± 20%. The attachment of the sensor significantly affects the accuracy of the results. For this reason, the AE sensors were fixed firmly to the surface of the rock using a high-degree vacuum adhesive. The amplitudes of the free amplifier and the main amplifier were amplified to 40 dB and 20 dB, respectively, because the crack initiation signal had a weak amplitude.

Figure 9 shows the AE signals from each AE sensor when the pencil lead break point (excitation point 1) was (x, y) = (125 mm, 0 mm). The AE signals from Ch.3–Ch.6 arrive earlier than the AE signals from Ch.1–Ch.2. This is because the pencil lead break point is farthest from sensor 1 and sensor 2, and between sensor 3 and sensor 6 (Figure 7).

In Figure 9, the defined initiating point (arrival time) has a high probability of error depending on the analyzer. As shown in Figure 10, arrival time occurs when the signal variance initiates variance according to the moving window size, which was defined as a result of the calculation of the raw AE signal with Equation (4) (the proposed moving window method). The estimated arrival time delay obtained using the proposed method is presented in Table 1.

Figure 11, Figure 12 and Figure 13 show the estimated source location results derived by the substitution of Table 1 in Equation (12). The colors represent the variance in AE wave velocity. The estimated location of the source is where velocity variance is the least. This is because the velocity variance becomes 0 where the scan source accords with the true source. In Table 2, we present the estimated results of the source location obtained using the above method. As shown in Table 2, the maximum error is 2.8 mm, so the proposed algorithm for arrival time delay and location estimation is valid.

The earlier results from the experiment using a rock specimen had a high signal-to-noise ratio and could not be used to verify detection capability, which is the biggest advantage of the proposed method. Instead, the experiment was carried out using artificial noise.

We supposed that the noise was white noise with a variance size of 340. The signal-to-noise ratios in sensors 1–6 were 0.025, 0.026, 0.015, 0.016, 0.016, and 0.012, respectively. These values are too low to define the initial point of the crack signal as directly as in Figure 14. However, as shown in Figure 15, we defined the initial point of the crack signal using the moving window method, and estimation was possible based on that initial point. Figure 16 shows the result of estimated source location calculated by arrival times from Figure 15. The estimated source location was (−124.7 mm, 0.5 mm) when the true source location was (−125 mm, 0 mm). This means that the proposed method is useful even when the signal-to-noise ratio is low.

## 6. Verification Test at In Situ Test

A field demonstration test was carried out in KURT (KAERI Underground Research Tunnel) in the KAERI (Korea Atomic Energy Research Institute) (Figure 17).

As shown in Figure 17, the wall of the tunnel, consisting of rocks, is finished with shotcrete, and the ground is concrete. The entire interior of the real tunnel is concrete; this research experiment was carried out on the ground. There were eight AE sensors placed in the tunnel, as shown in Figure 18. The distance between sensors was 0.5 m, and they were arranged in two lines. We excited several points (Figure 18a) using an impact hammer (Figure 18b) because we could not generate real cracks in the ground at several points. Excitation point 2, in particular, was used to investigate whether the sensors could calculate the crack location, even when the excitation occurred at the edge of the AE sensor array.

Figure 19 shows the results of the calculation with the moving window and the measured signals from each sensor when excitation point 1 was excited with an impact hammer. The conventional method would estimate the crack location by selecting the peak value of the crack signal as shown in Figure 19a. The time delay predicted using the conventional method and the proposed method is summarized in Table 3.

To compare the performance of the proposed method with that of the conventional method, the crack localization was estimated using Equation (12). Figure 20 shows the estimated impact locations using the calculated time delay from Table 3.

Figure 20a shows the results of source localization using the arrival times obtained by the conventional method, where ○ means the estimated location and X is the true source location. As shown in the results, the true location is (−0.25 m, 0.25 m) but the estimated location predicted by the conventional method is (0 m, 0.399 m). There is quite a large error of 0.291 m in distance. On the other hand, the estimated location by the proposed method is (−0.267 m, 0.248 m), as shown in Figure 20b, and this result has a distance error of 0.171 m, which is smaller than that of the conventional method.

To observe the performance of source localization when a source exists outside the eight-sensor array, the experiment was performed by impacting it at the location (1 m, −0.5 m), as shown in Figure 21. Similar to the previous results, it was found that the distance error predicted by the proposed method was smaller than the error estimated by the conventional method (presented in Table 4).

## 7. Conclusions

Crack signals from rock and concrete structures occur in conjunction with background noise from various causes. As a result, errors occur when estimating the crack location because the crack wave arrival time cannot be accurately determined. In this paper, by analyzing the crack signal and background noise characteristics, we were able to propose a concept of a ‘moving window’ to accurately determine the crack wave arrival time in a noisy environment. Experiments on a rock specimen and tunnel concrete showed that the proposed method is good at estimating crack locations.

In this paper, a method was suggested to increase the accuracy of fault location estimation using AE signals. It is expected that the proposed method can be applied to the ‘structural health monitoring’ of large structures to find defects and weaknesses, such as cracks. For example, it is expected that the proposed method could be utilized in various fields, such as the following:The effective detection of extremely early fatigue damage in structures, including aircraft.The safety assessment of civil structures, including steel and concrete bridges and dams.The diagnosis of in-process processes (e.g., welding monitoring).As a means of testing, evaluating, and identifying the mechanical properties and fracture mechanisms of materials.The detection and determination of the location of defects in pressure vessels, including nuclear reactors, etc.

## Figures and Tables

**Figure 1 sensors-24-06092-f001:**
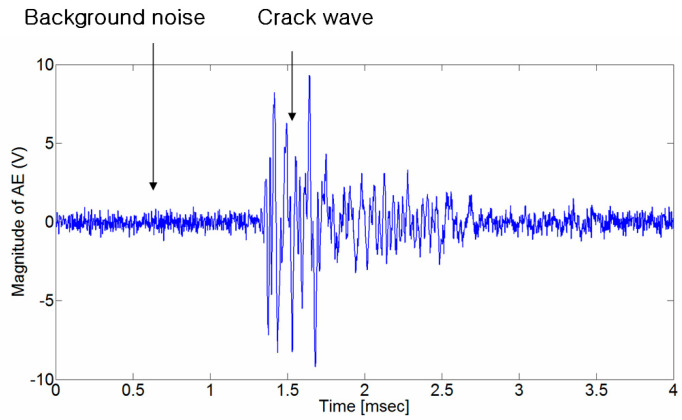
A crack wave generated in rock. The sampling frequency is 500 kHz and the number of data points is 4096. It is difficult to determine the arrival time of the p-wave in the presence of the ambient noise signal.

**Figure 2 sensors-24-06092-f002:**
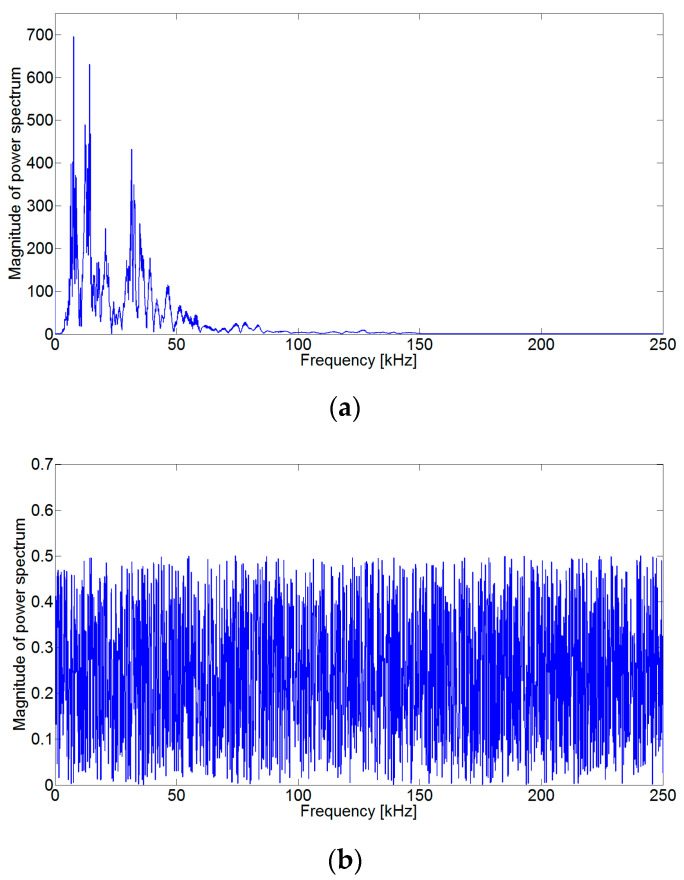
The frequency analysis results obtained from the Figure 1 signal. (**a**) Power spectrum for a crack wave and (**b**) power spectrum for background noise. In the frequency characteristics, we can observe that a crack wave has a narrow band, and the noise has a broad band.

**Figure 3 sensors-24-06092-f003:**
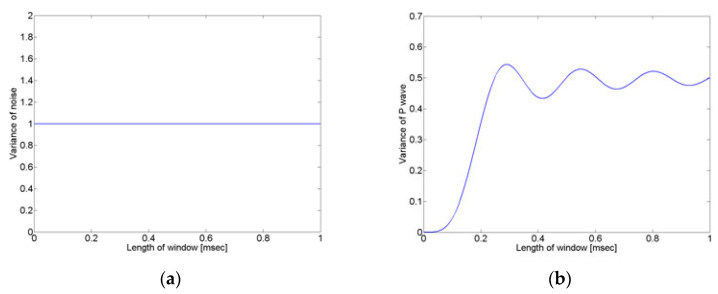
Signal variance according to window size for (**a**) noise signal and (**b**) crack wave.

**Figure 4 sensors-24-06092-f004:**
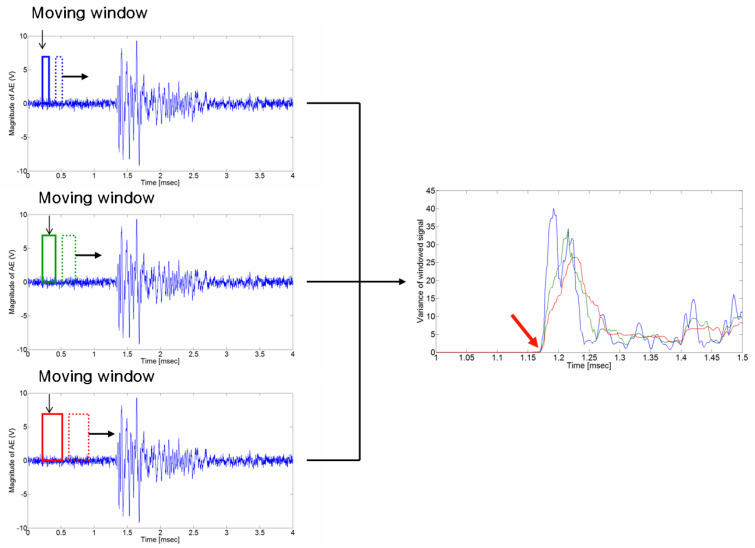
Proposed algorithm for determining the crack wave arrival time. While calculating the variance during window resizing, the point (the red arrow on the right) where the variances are different is the arrival time of the crack wave.

**Figure 5 sensors-24-06092-f005:**
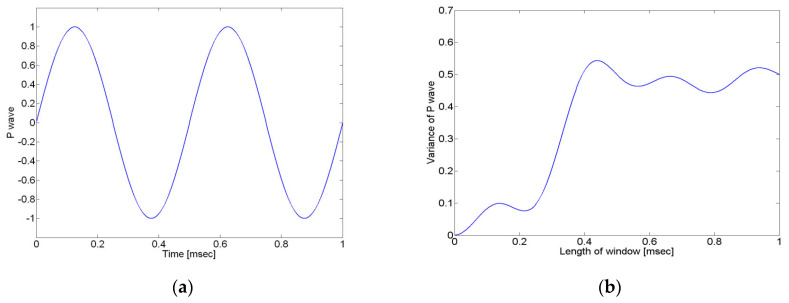
Results from applying the proposed method, assuming the crack wave is a sine wave. (**a**) Sine wave signal and (**b**) the variance in the sine wave using Equations (4) and (8). Because the rate of change in the variance is large within one wavelength of the signal, the window size should be larger than one wavelength.

**Figure 6 sensors-24-06092-f006:**
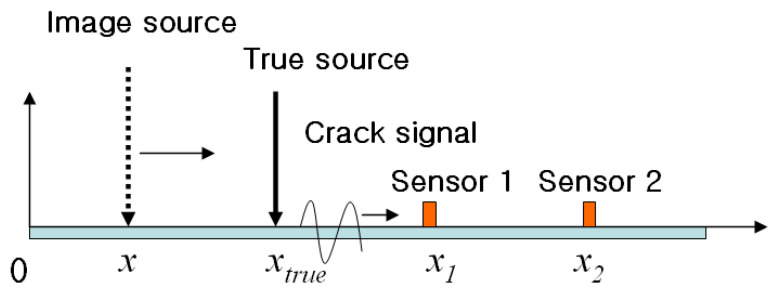
Concept explanation for source localization. If the velocities are calculated while scanning an image source, two velocities will match at the location of the true source.

**Figure 7 sensors-24-06092-f007:**
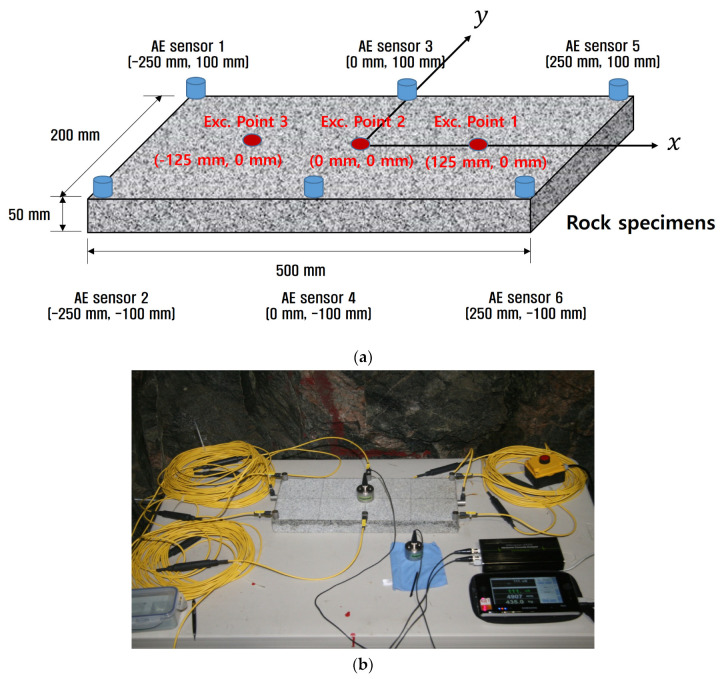
A schematic diagram of the pencil lead break experiment to test the accuracy of the method for source localization. (**a**) Location of the AE sensors and excitation points on the granite rock and (**b**) picture of the experimental setup.

**Figure 8 sensors-24-06092-f008:**
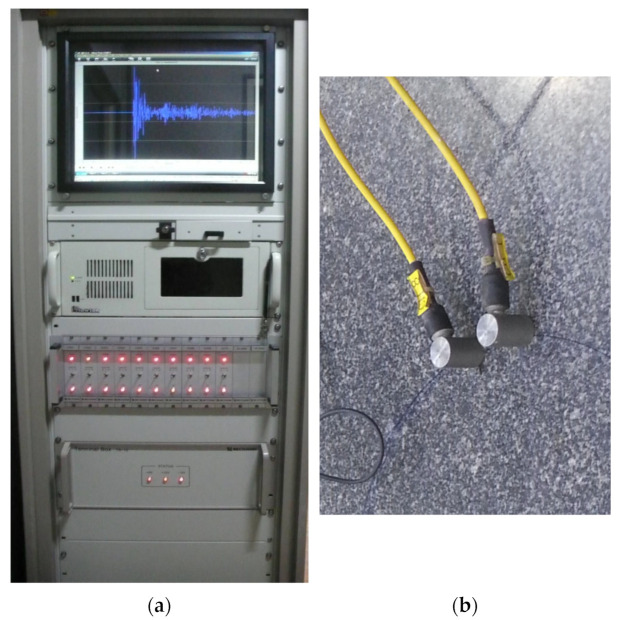
AE sensors and instruments used in this test. (**a**) AE-300 and (**b**) AE-603 SW-GA sensor.

**Figure 9 sensors-24-06092-f009:**
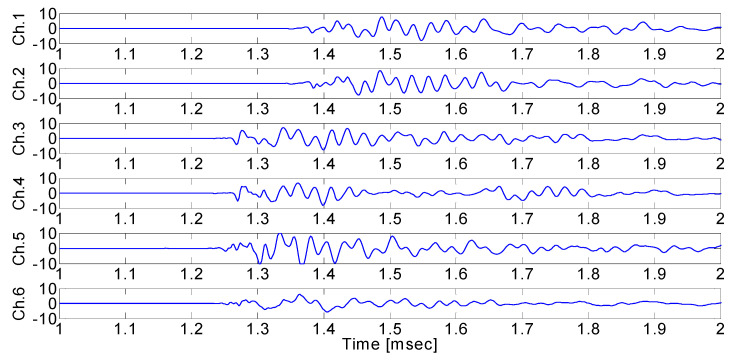
Measured AE signals when the pencil lead break test was performed at excitation point 1. Depending on the sensor locations, there is a difference in signal arrival delay.

**Figure 10 sensors-24-06092-f010:**
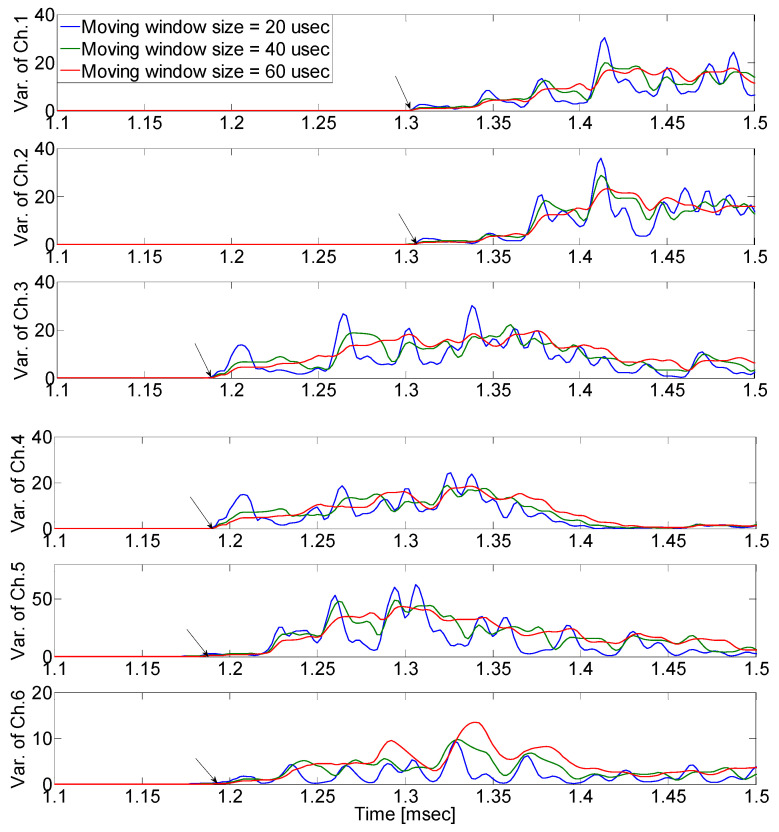
Experimental results using the proposed moving window method when the pencil lead break test was performed at excitation point 1. We can easily find the starting point (the black arrow) at which the variances in the measured signal change as the window size changes.

**Figure 11 sensors-24-06092-f011:**
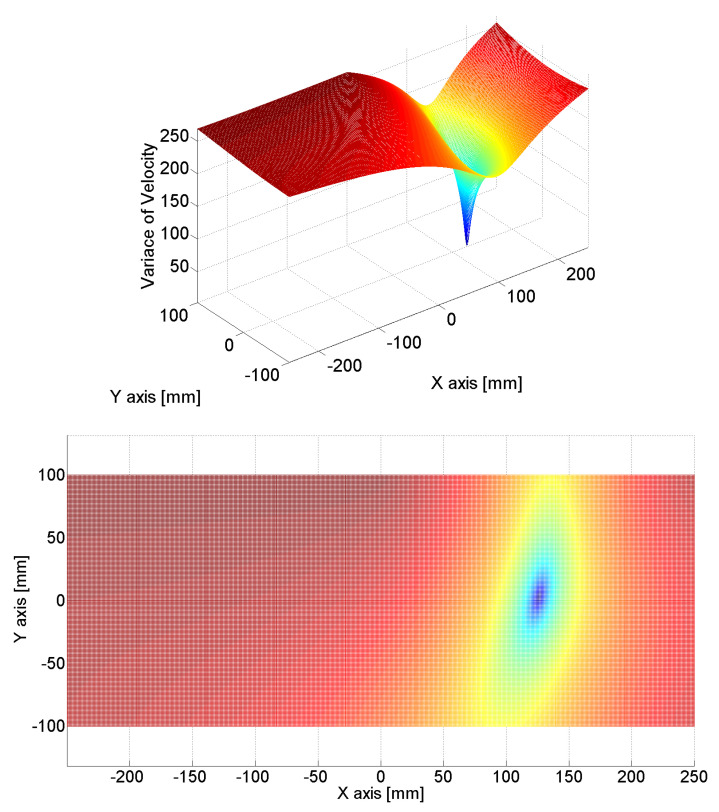
Source location was estimated by calculating the variance in velocities; the minimum value of variance indicates the source location. The estimated source location was (125.8 mm, 2.7 mm), while the true source location was (125 mm, 0 mm).

**Figure 12 sensors-24-06092-f012:**
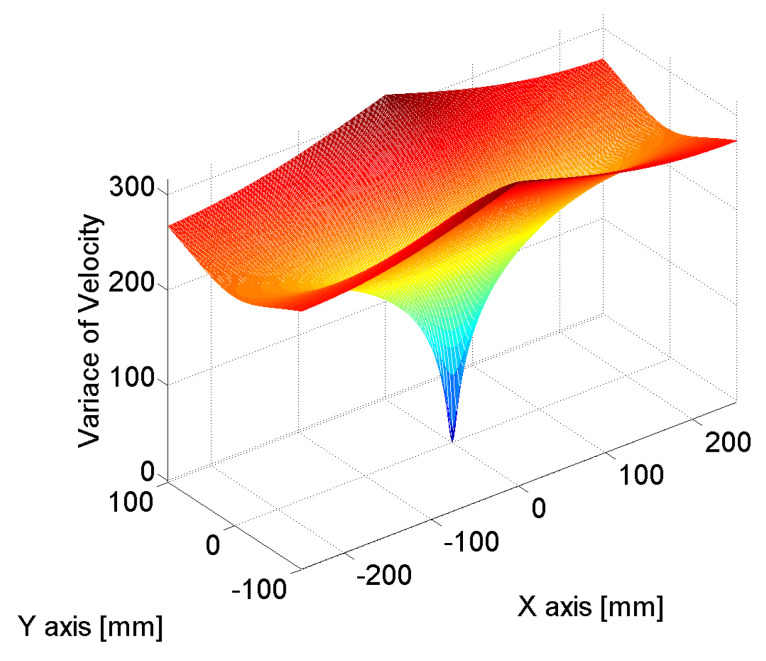
Source location was estimated by calculating the variance in velocities; the minimum value of variance indicates the source location. The estimated source location was (−0.5 mm, 0.5 mm), while the true source location was (0 mm, 0 mm).

**Figure 13 sensors-24-06092-f013:**
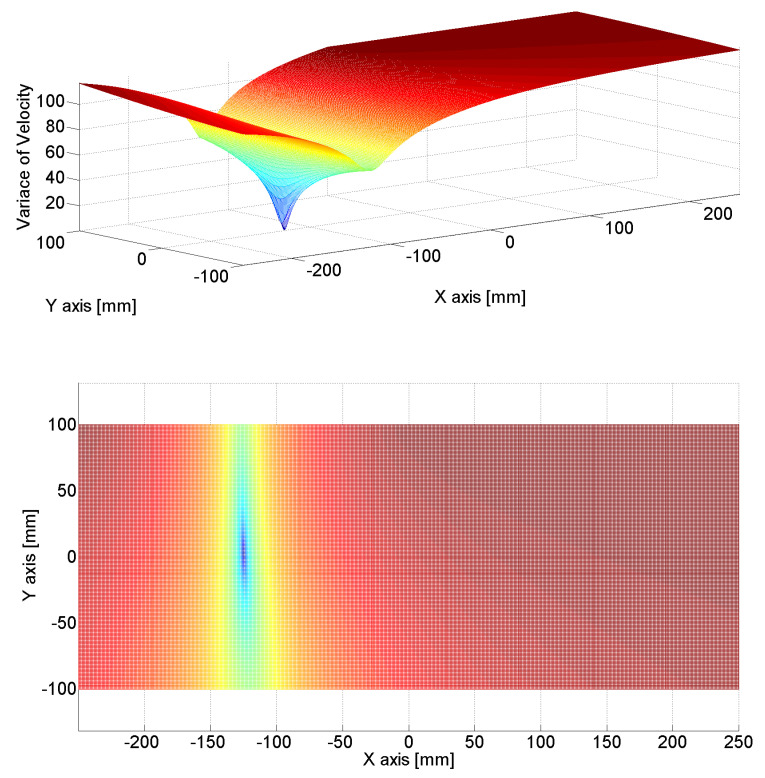
Source location was estimated by calculating the variance in velocities; the minimum value of variance indicates the source location. The estimated source location was (−124.7 mm, 1.5 mm), while the true source location was (−125 mm, 0 mm).

**Figure 14 sensors-24-06092-f014:**
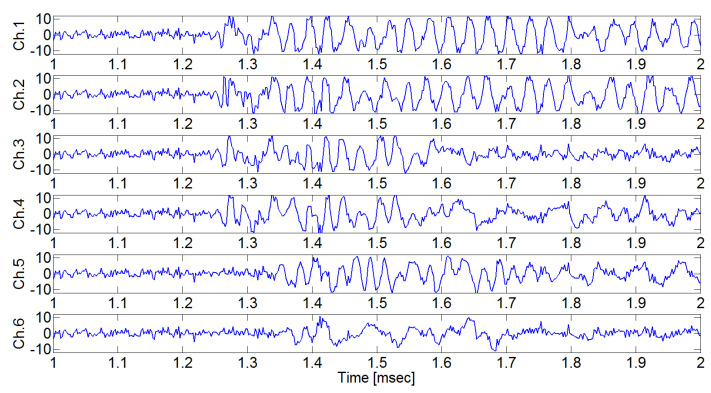
The signals were a mixture of artificial noise and Figure 9 signals when the pencil lead break test was performed at excitation point 1. Noise makes it more difficult to find the starting point of the crack signal.

**Figure 15 sensors-24-06092-f015:**
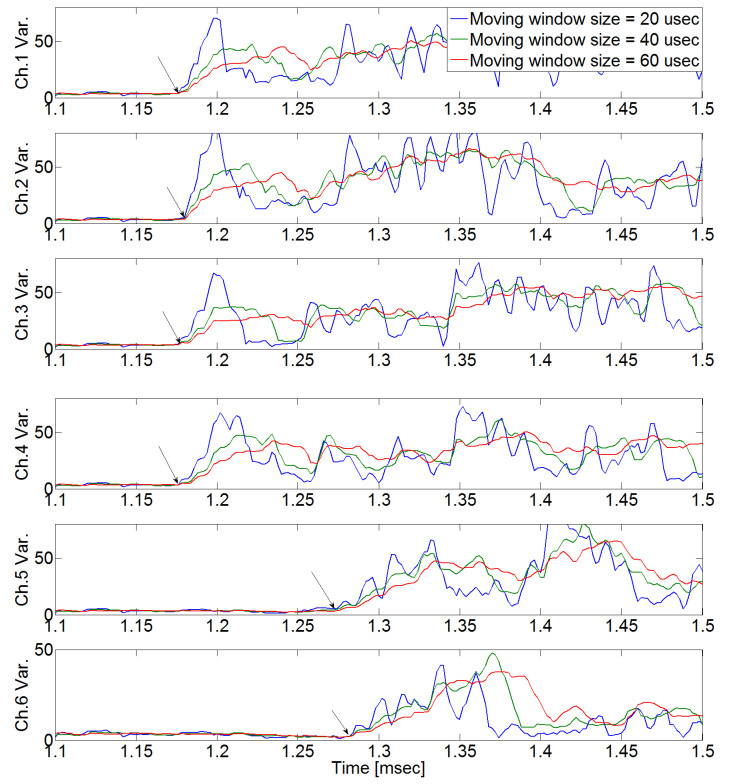
Experimental results in a noisy environment when the pencil lead break test was performed at excitation point 1. The black arrows mean the arrival times of crack wave. Applying the proposed method makes it easy to find the starting point of the crack signal even in a noisy environment.

**Figure 16 sensors-24-06092-f016:**
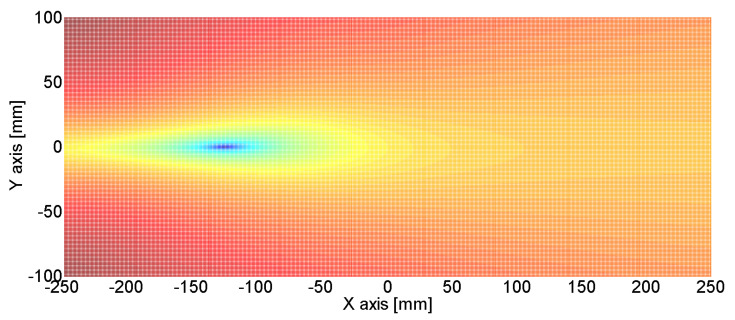
The source localization result. The estimated source location was (−124.7 mm, 0.5 mm) when the true source location was (−125 mm, 0 mm).

**Figure 17 sensors-24-06092-f017:**
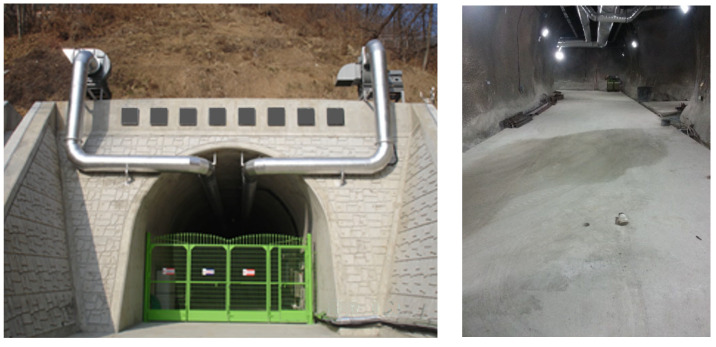
KURT (Korea Underground Research Tunnel) and research module.

**Figure 18 sensors-24-06092-f018:**
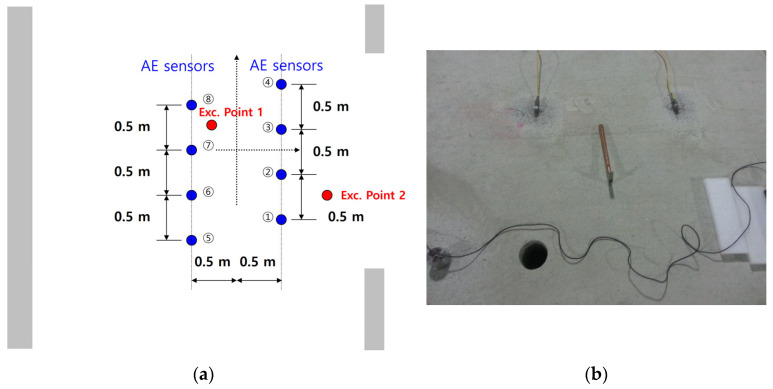
(**a**) Location of AE sensors and excitation points on the ground of the tunnel and (**b**) experimental pictures.

**Figure 19 sensors-24-06092-f019:**
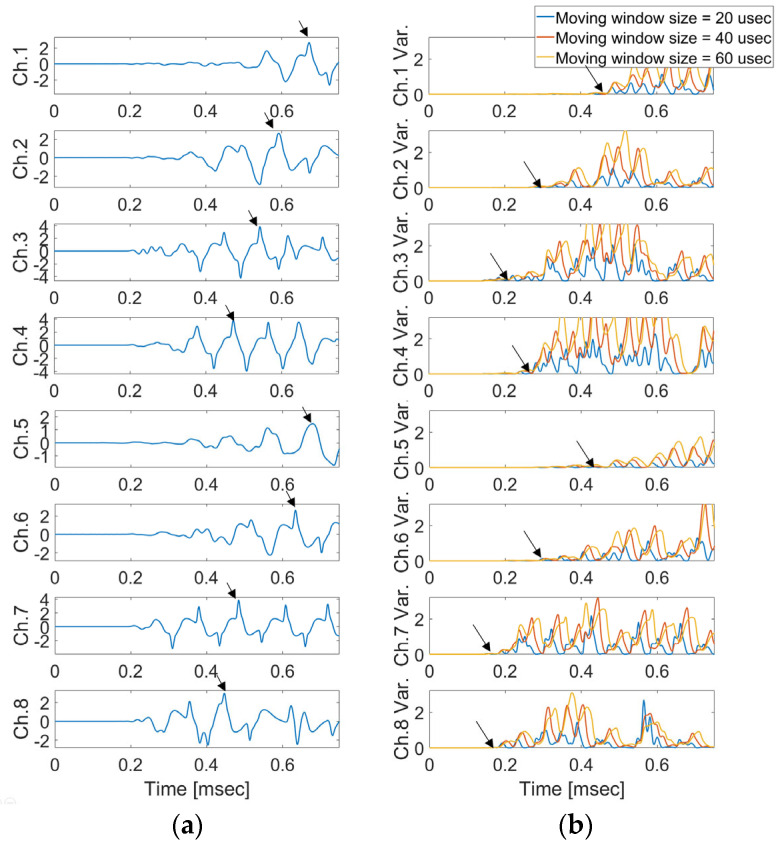
(**a**) Signal results from each sensor and (**b**) calculation of moving window when Exc. Point 1 was excited with an impact hammer. The black arrows mean the arrival time of impact wave.

**Figure 20 sensors-24-06092-f020:**
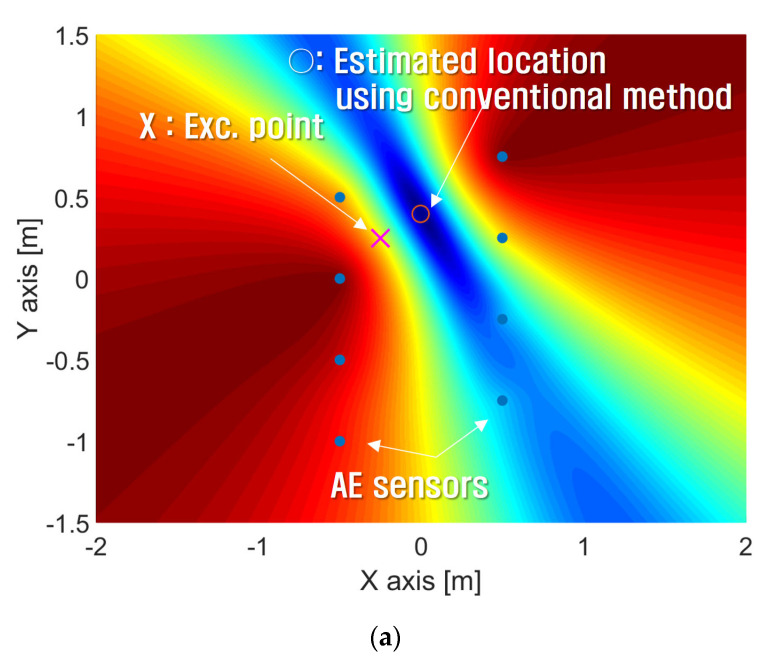
The result of the estimated impact location when Exc. Point 1 location is (x, y) = (−0.25 m, 0.25 m). (**a**) The estimated location applying the time delay obtained by the conventional method is (x, y) = (0 m, 0.399 m). (**b**) The estimated location applying the time delay obtained by the proposed method is (x, y) = (−0.267 m, 0.248 m). The blue dots represent sensor locations and X represents the excitation point.

**Figure 21 sensors-24-06092-f021:**
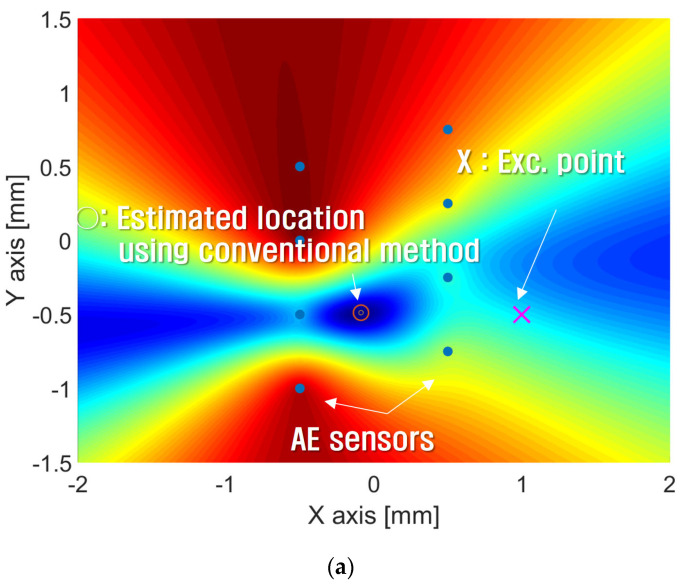
The result of the estimated impact location when Exc. Point 1 location is (x, y) = (1 m, −0.5 m). (**a**) The estimated location applying the time delay obtained by the conventional method is (x, y) = (−0.09 m, −0.49 m). (**b**) The estimated location applying the time delay obtained by the proposed method is (x, y) = (0.98 m, −0.52 m). The blue dots represent sensor locations and X represents the excitation point.

**Table 1 sensors-24-06092-t001:** Measured arrival time by using proposed method; arrival delay time of each sensor channel depending on source location.

	Exc. Point 1 (−125 mm, 0 mm)	Exc. Point 2 (0 mm, 0 mm)	Exc. Point 3 (125 mm, 0 mm)
	Arrival Time	Arrival Time	Arrival Time
Ch. 1	1.302 msec	1.232 msec	1.174 msec
Ch. 2	1.304 msec	1.236 msec	1.175 msec
Ch. 3	1.189 msec	1.170 msec	1.174 msec
Ch. 4	1.190 msec	1.170 msec	1.176 msec
Ch. 5	1.188 msec	1.234 msec	1.252 msec
Ch. 6	1.191 msec	1.235 msec	1.254 msec

**Table 2 sensors-24-06092-t002:** Source localization results.

	Source Locations (*x*, *y*)		
True source locations	Excitation point 3 (−125 mm, 0 mm)	Excitation point 2 (0 mm, 0 mm)	Excitation point 1 (125 mm, 0 mm)
Estimated source locations	(−125.8 mm, 2.7 mm)	(−0.5 mm, 0.5 mm)	(−124.7 mm, 1.5 mm)
Distance error btw true and estimated location	2.8 mm	0.7 mm	1.5 mm

**Table 3 sensors-24-06092-t003:** Estimated arrival time from eight sensors to estimate crack location.

Sensor	Estimated Arrival Time at Excitation Point 1		Estimated Arrival Time at Excitation Point 2	
	Conventional Method	Proposed Method	Conventional Method	Proposed Method
Sensor 1	0.672 msec	0.420 msec	0.668 msec	0.168 msec
Sensor 2	0.592 msec	0.268 msec	0.542 msec	0.256 msec
Sensor 3	0.542 msec	0.208 msec	0.628 msec	0.238 msec
Sensor 4	0.472 msec	0.232 msec	0.640 msec	0.308 msec
Sensor 5	0.788 msec	0.421 msec	0.670 msec	0.330 msec
Sensor 6	0.672 msec	0.292 msec	0.494 msec	0.382 msec
Sensor 7	0.484 msec	0.159 msec	0.518 msec	0.342 msec
Sensor 8	0.448 msec	0.160 msec	0.500 msec	0.396 msec

**Table 4 sensors-24-06092-t004:** Comparison of performance results for estimating the impact location using the proposed and conventional methods. The error means the distance between the true location and the estimated location.

Exc. Point	True Location	Conventional Method		Proposed Method	
		Estimated Location	Error	Estimated Location	Error
1	(−0.25 m, 0.25 m)	(0 m, 0.399 m)	0.291 m	(−0.267 m, 0.248 m)	0.171 m
2	(1.0 m, −0.5 m)	(−0.09 m, −0.49 m)	1.09 m	(0.98 m, −0.52 m)	0.028 m

## Data Availability

The original contributions presented in the study are included in the article, further inquiries can be directed to the corresponding author.

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
