# Peer review of "A Proposed Algorithm Based on Variance to Effectively Estimate Crack Source Localization in Solids"

_sensors, 2024, doi:10.3390/s24186092_

Round 1
Reviewer 1 Report
Comments and Suggestions for Authors
This paper introduces a novel algorithm for locating damage in solid materials, specifically cracks in rocks, by analyzing acoustic emissions (AE). The Time-of-arrival-differences (TOAD) algorithm is used to detect crack waves even with low signal-to-noise ratios, utilizing the distinct variances between the crack wave and noise across different window sizes. Successful testing with pencil lead breaks (PLB) on rock samples has demonstrated the algorithm's effectiveness in determining the precise origin of damage, which holds potential for real-time damage assessment in underground structures like tunnels.
The introduction consists of several short paragraphs. Please make sure each paragraph has sufficient length.
The literature review could benefit from adding references to similar publications in the area of acoustic emission monitoring. See for example https://doi.org/10.1016/j.engstruct.2019.109930
What types of differences do the authors expect between PLB and real AE?
Comments on the Quality of English Language
The language could benefit professional editorial services.
Author Response
Comments1:
This paper introduces a novel algorithm for locating damage in solid materials, specifically cracks in rocks, by analyzing acoustic emissions (AE). The Time-of-arrival-differences (TOAD) algorithm is used to detect crack waves even with low signal-to-noise ratios, utilizing the distinct variances between the crack wave and noise across different window sizes. Successful testing with pencil lead breaks (PLB) on rock samples has demonstrated the algorithm's effectiveness in determining the precise origin of damage, which holds potential for real-time damage assessment in underground structures like tunnels.
The introduction consists of several short paragraphs. Please make sure each paragraph has sufficient length.
The literature review could benefit from adding references to similar publications in the area of acoustic emission monitoring. See for example https://doi.org/10.1016/j.engstruct.2019.109930
Response 1:
We have revised the entire paper, including the introduction as your comment. I maked the modified part in blue.
Reviewer 2 Report
Comments and Suggestions for Authors
This paper proposes a variance-based time-of-arrival (TOA) picking algorithm that can accurately determine the crack source location in solids under noisy environments. By employing a moving window technique, the algorithm effectively identifies the arrival time of crack waves even in low signal-to-noise ratio (SNR) conditions and estimates the precise crack location by minimizing the velocity variance. Experimental results show that the algorithm performs well in monitoring rock samples and underground facilities, making it suitable for real-time health monitoring of tunnels or other underground structures. The article would benefit from the following revisions before publication:
1. The paper provides insufficient details about the pencil lead break and in-situ experiments. More information should be included regarding the experimental setup (e.g., sensor arrangement, environmental conditions) to enhance the reader's understanding of the validity of the results.
2. While the main formulas are presented, some derivations are unclear. Additional intermediate steps and explanations should be provided to allow readers to follow the theoretical foundation of the algorithm more easily.
3. When presenting experimental results, clearer graphs and annotations are recommended. Specifically, for comparisons of different signal-to-noise characteristics, include additional markers or color coding to make the key points more apparent.
4. The discussion section is relatively brief and could be expanded to explore potential applications of the algorithm in other fields or with different materials. Additionally, a comparison with existing methods and a description of future research directions would be beneficial.
5. The article includes only thirteen references, lacking sufficient coverage of related work in crack localization. Several key papers should be cited, such as:
Joint Inversion of AE/MS Sources and Velocity with Full Measurements and Residual Estimation. Rock Mech Rock Eng 2024 1–16.
Adding references will strengthen the literature review and demonstrate a comprehensive understanding of the field.
Author Response
Comments2:
This paper proposes a variance-based time-of-arrival (TOA) picking algorithm that can accurately determine the crack source location in solids under noisy environments. By employing a moving window technique, the algorithm effectively identifies the arrival time of crack waves even in low signal-to-noise ratio (SNR) conditions and estimates the precise crack location by minimizing the velocity variance. Experimental results show that the algorithm performs well in monitoring rock samples and underground facilities, making it suitable for real-time health monitoring of tunnels or other underground structures. The article would benefit from the following revisions before publication:
1. The paper provides insufficient details about the pencil lead break and in-situ experiments. More information should be included regarding the experimental setup (e.g., sensor arrangement, environmental conditions) to enhance the reader's understanding of the validity of the results.
2. While the main formulas are presented, some derivations are unclear. Additional intermediate steps and explanations should be provided to allow readers to follow the theoretical foundation of the algorithm more easily.
3. When presenting experimental results, clearer graphs and annotations are recommended. Specifically, for comparisons of different signal-to-noise characteristics, include additional markers or color coding to make the key points more apparent.
4. The discussion section is relatively brief and could be expanded to explore potential applications of the algorithm in other fields or with different materials. Additionally, a comparison with existing methods and a description of future research directions would be beneficial.
5. The article includes only thirteen references, lacking sufficient coverage of related work in crack localization. Several key papers should be cited, such as:
Joint Inversion of AE/MS Sources and Velocity with Full Measurements and Residual Estimation. Rock Mech Rock Eng 2024 1–16.
Adding references will strengthen the literature review and demonstrate a comprehensive understanding of the field.
Response 2:
We have revised the entire paper as your comment. I marked the modified part in blue.
- We have added a detailed descriptions and figures
- We have added an additional steps at 8~9 pages.
- We have added a detailed descriptions and figures at entire paper.
- We have added applications and comparison with conventional method.
- We have added references.